# Agreement between Handheld and Standard Echocardiography for Diagnosis of Latent Rheumatic Heart Disease in Brazilian Schoolchildren from High-Prevalence Settings (Agreement between Screening and Standard Echo for RHD) [note 1]

**DOI:** 10.3390/diagnostics14040392

**Published:** 2024-02-11

**Authors:** Marina G. Diniz, Lucas L. Fraga, Maria Carmo P. Nunes, Kaciane K. B. Oliveira, Ingred Beatriz Amaral, Luz Marina T. Chavez, Luiza Haikal de Paula, Beatriz C. Haiashi, Alexandre M. Ferreira, Mauro Henrique A. Silva, Jéssica Elvira M. Veloso, Cássia Aparecida Silva, Fernanda A. Gelape, Luiza P. A. Santos, Arthur M. Amaral, Cecília T. Coelho, Lucas C. Diamante, Juliane S. Correia, Zilda Maria A. Meira, Antonio Luiz P. Ribeiro, Alison M. Spaziani, Craig Sable, Bruno R. Nascimento

**Affiliations:** 1Serviço de Cardiologia e Cirurgia Cardiovascular e Centro de Telessaúde do Hospital das Clínicas da Universidade Federal de Minas Gerais, Belo Horizonte 30130-100, MG, Brazil; marinagd.diniz@gmail.com (M.G.D.); mcarmo@waymail.com.br (M.C.P.N.); kacianebruno@hotmail.com (K.K.B.O.); ingred.bh.beatriz@gmail.com (I.B.A.); luztacuri78@gmail.com (L.M.T.C.); luizahaikaldepaula@gmail.com (L.H.d.P.); beatrizhaiashi@gmail.com (B.C.H.); alexandredemelo29@gmail.com (A.M.F.); maurohenrique1209@gmail.com (M.H.A.S.); jessicaepmachado@gmail.com (J.E.M.V.); cassiaapsilva1@gmail.com (C.A.S.); ceciliatcoelho@hotmail.com (C.T.C.); lucasdiamante02@gmail.com (L.C.D.); jullianesc1@gmail.com (J.S.C.); zilda.m.a.meira@gmail.com (Z.M.A.M.); tom1963br@yahoo.com.br (A.L.P.R.); 2Departamento de Clínica Médica, Faculdade de Medicina da Universidade Federal de Minas Gerais, Belo Horizonte 30130-100, MG, Brazil; 3Curso de Medicina, Faculdade de Ciências Médicas de Minas Gerais, Belo Horizonte 30130-110, MG, Brazil; fernandagelape@hotmail.com (F.A.G.); luizaapereirasantos@gmail.com (L.P.A.S.); 4Departamento de Medicina, Universidade Federal de Ouro Preto, Ouro Preto 35400-000, MG, Brazil; arthur29maia@gmail.com; 5Cardiology, Children’s National Health System, Washington, DC 20010, USA; aspaziani@childrensnational.org (A.M.S.); csable@childrensnational.org (C.S.); 6Serviço de Hemodinâmica, Hospital Madre Teresa, Belo Horizonte 30441-070, MG, Brazil

**Keywords:** rheumatic heart disease, screening, echocardiography, accuracy, agreement

## Abstract

**Introduction**: Handheld echocardiography (echo) is the tool of choice for rheumatic heart disease (RHD) screening. We aimed to assess the agreement between screening and standard echo for latent RHD diagnosis in schoolchildren from an endemic setting. **Methods:** Over 14 months, 3 nonphysicians used handheld machines and the 2012 WHF Criteria to determine RHD prevalence in consented schoolchildren from Brazilian low-income public schools. Studies were interpreted by telemedicine by 3 experts (Brazil, US). RHD-positive children (borderline/definite) and those with congenital heart disease (CHD) were referred for standard echo, acquired and interpreted by a cardiologist. Agreement between screening and standard echo, by WHF subgroups, was assessed. **Results:** 1390 students were screened in 6 schools, with 110 (7.9%, 95% CI 6.5–9.5) being screen positive (14 ± 2 years, 72% women). Among 16 cases initially diagnosed as definite RHD, 11 (69%) were confirmed, 4 (25%) reclassified to borderline, and 1 to normal. Among 79 cases flagged as borderline RHD, 19 (24%) were confirmed, 50 (63%) reclassified to normal, 8 (10%) reclassified as definite RHD, and 2 had mild CHD. Considering the 4 diagnostic categories, *kappa* was 0.18. In patients with borderline RHD reclassified to non-RHD, the most frequent WHF criterion was B (isolated mitral regurgitation, 64%), followed by A (2 mitral valve morphological features, 31%). In 1 patient with definite RHD reclassified to normal, the WHF criterion was D (borderline RHD in aortic and mitral valves). After standard echo, RHD prevalence was 3.2% (95% CI 2.3–4.2). **Conclusions:** Although practical, RHD screening with handheld devices tends to overestimate prevalence.

## 1. Introduction

Acute rheumatic fever (ARF) and its late sequelae, rheumatic heart disease (RHD), represent conditions within the spectrum of acquired autoimmune diseases, manifesting comprehensively and impactfully, especially in the cardiovascular system [1,2]. ARF emerges as an inflammatory response triggered by infection with Group A Streptococcus, manifesting through multifaceted symptoms encompassing arthritis, arthralgia, chorea, subcutaneous nodules, and carditis [3,4]. It is noteworthy that approximately 50–65% of individuals diagnosed with ARF exhibit cardiac involvement, evidenced by inflammation and fibrosis of valve leaflets, with the mitral valve being the most frequently affected [5].

Thus, chronic RHD becomes a leading cause of cardiovascular disease in children and young adults, especially in underdeveloped and developing countries [6,7]. The global prevalence of RHD reached 40.5 million cases in 2019, according to data from the Global Burden of Disease (GBD) study, resulting in approximately 306,000 deaths annually [6]. Remarkably, the majority of these cases are concentrated in populations of countries with limited resources, especially in sub-Saharan African and Latin American nations, including Brazil, where prevalence has proven to be alarmingly high for both latent and clinical stages [6,8,9].

It is concerning to note that most patients diagnosed with RHD do not have a clear history of ARF when seeking medical attention, indicating a significant underdiagnosis of this condition [10,11]. Additionally, a considerable portion of fatalities could be prevented or delayed through early detection of the disease [11]. In this context, portable echocardiographic (echo) screening emerged in the past decades as a safe and effective approach for the early detection of cardiac alterations, significantly contributing to the reduction of adverse outcomes [12,13].

In 2012, the World Heart Federation (WHF) proposed the use of portable echocardiographic devices for RHD screening in schoolchildren, especially those in endemic areas [14]. In the same initiative, the WHF established diagnostic criteria for latent RHD in two categories: definite and borderline [14]. Recently, the new 2023 WHF guidelines went deeper into this concept, including a more comprehensive and clinically meaningful classification of latent RHD, with risk stratification that includes a score based on a set of echo variables [15,16].

In Brazil, where RHD is implicated in nearly 50% of valve heart surgeries within the public health system [17,18], the first large-scale screening program for RHD was initiated only in 2014 [19]. This effort stemmed from an international collaboration involving a federal public university in southeast Brazil and various institutions, particularly the Children’s National Health System in Washington, DC, USA. Initially, a screening program was established in public schools based on health education, followed by echocardiographic screening performed by nonphysicians [20]. This was supported by a robust telemedicine cloud system, allowing remote interpretation by experts in Brazil and abroad [21]. This initiative was groundbreaking and contentious, as the use of ultrasound imaging, even for screening with simplified protocols, was not permitted outside of research according to Brazilian medical regulations. Following this initial phase, which revealed a prevalence of latent RHD as high as 4.5%, the RHD project expanded to primary care [19,20].

However, the agreement for specific WHF variables between screening echo acquired with ultraportable devices by nonphysicians and the final diagnosis by an expert in fully functional machines still needs additional investigation [22]. This study aimed to evaluate the agreement between screening echocardiography performed by non-specialists with handheld devices and conventional echocardiography carried out by experienced cardiologists in diagnosing latent RHD in high-endemicity settings in Brazil.

## 2. Methods

The procedures and methods of this study will be made available for replication upon reasonable request directed to the corresponding author. The PROVAR+ (*Programa de Rastreamento da VAlvopatia Reumática e Outras Doenças Cardiovasculares*) initiative represents a collaborative research and healthcare effort between the Universidade Federal de Minas Gerais, through the Telehealth Network of Minas Gerais, Brazil, and the Children’s National Health System in Brazil, USA [21]. Cross-sectional data utilized in this sub-study were collected between 2022 and 2023. Institutional review board (IRB) approvals were obtained from the Children’s National Health Hospital and the Universidade Federal de Minas Gerais under the number CAAE 24636713.9.0000.5149, as well as from local Boards of Health and Education.

The selection of schools in low-income areas of metropolitan Belo Horizonte was based on socioeconomic criteria, including the Human Development Index (HDI) and local indicators of health vulnerability, guided by local regulatory authorities. According to the inclusion criteria, all students from 11 to 18 years of age (based on the highest prevalence range of previous studies) in the selected schools, asymptomatic and without previous history of ARF or RHD, were eligible for participation. There were no restrictions related to nutritional or pubertal status. To improve adherence, informational letters and educational flipcharts (Figure 1) were presented to students and school staff during educational sessions, and copies were sent home. Signed parental informed consent and assent forms were required prior to enrollment for those <18 years of age, and a self-informed consent form was mandatory for those ≥18 years of age (Figure 2).

A team comprising one nurse research coordinator, one research nurse, and one technician conducted school screening with the support of medical students and local health agents. Training about RHD principles and basic imaging acquisition in a prespecified 7-view protocol included an online educational module and a minimum 32-h hands-on training, supervised by a cardiologist (MN) in the University’s Echocardiography Lab, over a 12-week period.

Consentedstudents provided demographic and socioeconomic information through a structured questionnaire and underwent a simplified ultrasound protocol focused on the left-sided cardiac valves using Lumify^®^ (Phillips, Eindhoven, The Netherlands) and VSCAN Extend^®^ (GE Healthcare, Chicago, IL, USA) devices. Cloud computing solutions (SigTel^®^, Universidade Federal de Minas Gerais, Brazil, and Tricefy^®^, Trice Imaging, San Diego, CA, USA) facilitated the storage and analysis of images. Telemedicine analysis by cardiologists in Brazil (MCN and JM) and the US (CS) classified children as “normal”, “borderline RHD”, “definite RHD”, or “other” based on the 2012 World Heart Federation criteria for individuals aged ≤20 years (Table 1, Figure 2), modified for the absence of spectral Doppler. Color regurgitant jets and morphologic signs of RHD were applied. Discrepancies were resolved after discussion, ensuring consensus diagnosis.

Positive cases (borderline, definite, and other) were referred to the UFMG University Hospital for follow-up standard echocardiography (Vivid IQ^®^, GE Healthcare), acquired and interpreted by 3 RHD experts (MCN, JM, and LM), and clinical evaluation (ZM). Depending on the local logistics of each school, aiming to improve participation, sometimes the follow-up standard echo was carried out in the school with the same devices and by the same certified cardiologists. Care decisions were made at the discretion of the experienced RHD cardiology team, including secondary prophylaxis with 3-weekly penicillin injections for selected definite cases and biannual medical surveillance for borderline RHD. All families received letters about the diagnosis and were counseled about the disease and the need for medical care [20].

## 3. Statistical Analysis

The data were systematically entered into the RedCap^®^ online database [23], and statistical analysis employed the SPSS^®^ software version 23.0 for Mac OSX (IBM, Armonk, NY, USA). The variable distribution was assessed using the Shapiro-Wilk test. Continuous variables were expressed as mean ± standard deviation or median and interquartile range when appropriate. The categorical variables were presented as absolute values and percentages. The comparisons between groups used the Student *t*-test or Mann-Whitney test. The categorical variables were compared using Fisher’s exact test, when appropriate. The screening and follow-up echocardiograms were compared in terms of the final diagnostic category and based on each set of criteria (diagnostic group) of the 2012 WHF guidelines (Table 1). The overall *kappa* statistic and the % agreement for each variable were reported, as well as sensitivity, specificity, and negative and positive predictive values of screening for the presence of any latent RHD. A two-tailed significance level of 0.05 was considered statistically significant for comparisons.

## 4. Results

A total of 1390 students underwent screening in six schools, of which 110 (7.9%, 95% CI 6.5–9.5) were identified as positive in screening (latent RHD or other relevant heart disease, namely congenital heart disease) and referred for standard follow-up echocardiography. The mean age of this subset was 14 ± 2 years (range 11 to 19), with 80 (72%) being female. The median household was 4.0 (IQR 3.0–5.0) inhabitants. A total of 76 (69%) standard confirmatory echocardiograms were performed directly at schools.

Overall, among the 16 cases initially diagnosed as definite RHD (1.2% of the sample), 11 (69%) were confirmed, 4 (25%) were reclassified to borderline RHD, and 1 to normal. Of the 79 cases classified as borderline RHD (5.7% of the sample), 19 (24%) were confirmed, 50 (63%) were reclassified as normal, 8 (10%) reclassified as definite RHD, and 2 presented mild congenital heart disease. Of the 11 cases of congenital heart disease identified in the screening, 4 were confirmed, 5 were reclassified as normal, and 2 reclassified as RHD (Table 2 and Table 3). The 6 cases with the final diagnosis of mild congenital heart disease were classified as mitral valve prolapse (N = 3), bicuspid aortic valve (N = 1), and atrial septal defects (N = 2). When considering the four diagnostic categories, the overall *kappa* coefficient was 0.18 for this sample and *kappa* = 0.13 for any latent RHD, resulting in a sensitivity of 95.5% (95% CI 84.3–99.4), specificity of 19.7% (95% CI 10.9–31.3), negative predictive value of 98.9% (95% CI 95.6–99.7), and positive predictive value of 5.3% (95% CI 4.7–6.0) in a prevalence scenario around 4.5% [20]. Detailed diagnostic test indicators [24], according to RHD category (all screen-positive and definite RHD), are provided in Table 4.

Among patients initially diagnosed as borderline RHD and subsequently reclassified as non-RHD (normal or other heart disease), the most frequent WHF criterion was B (isolated mitral regurgitation): 64%, followed by A (two morphological features of the mitral valve): 31%. Thus, the main cause for misclassification in this category was overestimation of mitral regurgitation jets. In the single patient with definite RHD reclassified as normal, the WHF criterion was D (borderline RHD in the aortic and mitral valves). After standard echocardiography, the prevalence of RHD was 3.2% (95% CI 2.3–4.2), with 1.8% being borderline RHD and 1.4% being definite RHD (Table 2 and Table 3).

## 5. Discussion

In our study, from a large screening program in a high-prevalence Latin American setting, there was a considerable degree of disagreement between screening echo task-shifted to nonphysicians, acquired with handheld devices, and standard fully functional exams performed by cardiologists. The disagreement was especially significant for the borderline category, and associated with overestimation of mitral regurgitation. For the definite category, however, agreement was higher, considering the association of morphological and functional findings. Our results point towards the need for refinement of education and training for screening personnel. They also reinforce the importance of a comprehensive risk stratification combining high-risk echo features into prediction scores [16] and the use of sociodemographic and clinical data.

The disparity in access to health services is a global issue that directly impacts the prevalence and severity of various diseases, including ARF and its chronic cardiac complications [25,26]. In regions with limited resources, such as Brazil and Latin America in general, the scarcity of access to appropriate medical care contributes to underdiagnosis and the silent progression of these conditions [27]. The introduction of portable and handheld echocardiography devices for RHD screening represents a significant innovation in the early detection of cardiac structural abnormalities, especially in contexts where access to the healthcare system is limited and challenging [28,29]. These devices emerge as an affordable, rapid, and effective approach for screening at-risk populations, potentially reducing disparities in diagnosis and lowering complication rates and unfavorable outcomes [30,31].

Despite resistance from medical councils in Brazil regarding the task-shifting of simplified handheld ultrasound protocols to nonphysicians, this strategy has proven successful for local RHD-related research purposes under the scrutiny of institutional review boards [20,21]. Telemedicine plays a crucial role in this scenario, facilitating well-structured storage and collaborative reading, along with the recent incorporation of artificial intelligence features [21,32]. The accuracy of professionals with diverse backgrounds in acquiring simple 7-view echo protocols and identifying abnormalities as screen-positive has been demonstrated [33]. The prevalence of subclinical RHD was assessed for the first time, revealing alarming rates comparable to some African countries [20].

The program was subsequently expanded to primary care, emphasizing early diagnosis of heart disease in adults and the elderly to rationalize referrals. The implementation was successful and well-received, revealing a high prevalence of undiagnosed heart disease, particularly among those in long waiting queues for cardiology appointments and specialized tests such as standard echo [19]. More recently, a risk score has been developed, integrating clinical data, tele-ECG signals, and screening echo variables to predict the presence of major structural heart disease. This holds promise as a tool for resource-limited regions. Thus, the past decade has witnessed significant advances in RHD research in Brazil, and the assessment of agreement between key tests for diagnosing subclinical stages, and consequently early referral for care, is crucial for refining screening strategies.

Other previous studies with nonspecialist professionals had demonstrated the capacity for adequate imaging acquisition and significant—although variable—correlation between screening echocardiography performed by nonspecialists and conventional echocardiography performed by specialists for diagnosing RHD [12,15,30]. This screening strategy is recognized as valid despite exhibiting higher sensitivity than specificity [34,35,36]. Early detection through echocardiographic screening allows for regular clinical monitoring and the implementation of secondary prophylaxis involving the use of G benzathine penicillin to prevent the progression of cardiovascular damage and eventual clinical deterioration [37].

Despite the evident advantages and benefits, our study identified a tendency of portable echocardiography to overestimate the prevalence of RHD, especially in cases initially classified as borderline RHD, highly associated with overestimation of regurgitant jets [34,35,36]. This observation reinforces the need for careful analysis and interpretation of data and images obtained during ultraportable echocardiography screening [16]. Additionally, the *kappa* coefficient stands out as a crucial element for evaluating the results, providing a deeper analysis of the agreement between screening using simplified protocols and task-shifting and standard echocardiography. This indicator reveals the consistency and reliability of the diagnostic approach applied, offering valuable insights into the effectiveness of screening for identifying cases of RHD compared to the gold standard represented by standard echocardiography. The agreement between these diagnostic modalities provides valuable insights into the validity and clinical utility of ultraportable echocardiography. Although the modality is highlighted by preexisting screening studies as a potentially effective screening tool for early detection of cardiac abnormalities in resource-limited high-prevalence scenarios, our data showed a much higher sensitivity—aligned with the requirements for a triage test—but lower specificity compared to available pooled data [28,34,35,36]. Among several possible causes, the heterogeneity of the training background of sonographers—a modality not yet allowed in Brazil outside research—and the limited features of ultrasound devices, in addition to technical challenges for acquiring images at schools, may have accounted for the disagreement.

Other studies on the disagreement between echocardiography performed by nonspecialists and that deployed by echocardiographers have yielded similar results in terms of overestimating mitral regurgitation, notably in borderline and mild definite cases, and misidentification of mild valvular morphological findings [22,33,36,38]. Similarly to our observations, authors attribute the discordance to differences in training and expertise between specialist and nonspecialist professionals, as well as the less powerful nature of portable devices compared to traditional ones, leading to limitations such as penetration, harmonization, and image resolution [22,39]. In addition, only more recent high-end handheld devices incorporated spectral Doppler capabilities. Despite prior studies suggesting the need for longer, specific, and practical training for nonspecialist professionals, including those with different backgrounds (nursing, nursing technician, or physiotherapy) [33,40], the impact of such training on screening outcomes remains unclear, especially on a larger scale.

In 2023, the WHF released new guidelines for diagnosing RHD, introducing changes to the diagnostic flow and revising patient classification and risk stratification [15]. In this update, the categories definitive, latent, and borderline were replaced by stages: positive screening A (minimal RHD echo criteria), B (mild RHD), C (advanced RHD), and D (advanced RHD with clinical complications) [15]. The new proposal considers as reasonable in special cases initiation of the prescription of secondary prophylaxis for RHD in screen-positive patients while awaiting confirmation through confirmatory echocardiography, depending on the propensity for progression as assessed by a validated echocardiographic score [16]. This approach may be more applicable in countries with limited access to health services where the interval between screening echocardiography and conventional diagnosis may be prolonged. However, it is important to note that the criteria for classification as positive screening include the presence of pathological mitral regurgitation, pathological aortic regurgitation, and/or reduced mobility and opening of the mitral valve [15,16]. Thus, echocardiographic findings from screening, stratified by risk, remain the cornerstone of this diagnostic–therapeutic flow for RHD [41]. However, our study suggests that such findings may be overestimated by ultraportable echocardiography, highlighting the importance of constant surveillance, rigorous training and quality assurance, and continuous protocol development to maximize the clinical utility of this screening strategy.

## 6. Limitations and Strengths

Our study has several limitations, mostly related to technical issues and the accuracy of screening echocardiography provided through ultraportable devices operated by nonphysicians. At first, the scanner who acquired the screening images had limited training, having completed online interactive modules and an average of 32 hours of hands-on training. Although one research nurse had been involved in the project for over 8 years, others were recently incorporated. This could have made the imaging quality heterogeneous, although the quality-assurance approaches applied tend to smoothen such discrepancies. Second, all students were included in school screening based on the parental granting of informed consent, without stratified sampling procedures. However, the study was not aimed at estimating overall prevalence, and the results cannot be extrapolated to the entire Brazilian population, nor to other screening programs. Third, specific features of the handheld devices applied may have contributed to heterogeneity and disagreement, markedly the absence of spectral Doppler and the limited resolution and penetration, especially for individuals with challenging anatomies. On the other hand, the long-term utilization of similar machines by the team, and the recent incorporation of newer devices, help overcome these technical drawbacks. Fourth, for logistic and budgetary reasons, false-negative screening studies were not verified, as only screen-positive children were referred for standard echo, and this may limit more definite conclusions about accuracy. However, this approach is aligned with current RHD diagnostic guidelines, as handheld screening has proven to be much more sensitive than specific. Fifth, due to the pragmatic study protocol, breakdown of the analyses by variables such as as pubertal and nutritional status, or by familial history of RHD, was not possible. Finally, this study was designed to look for agreement based on the 2012 WHF diagnostic criteria, and data collection was not adequate to apply the newly available 2023 criteria. Despite such limitations, our study provides important insights about the accuracy and clinical utility of echocardiographic screening for RHD, reinforcing the need for applying risk stratification as suggested by prospective studies and incorporated by the recent 2023 WHF guidelines. Moreover, the findings point out variables that must be more carefully interpreted, notably by experienced physicians with previous training in RHD diagnosis and management.

## 7. Conclusions

Although practical and potentially cost-saving, RHD screening with handheld devices tends to overestimate prevalence, especially concerning the overestimation of valve regurgitation. Even in endemic settings, continuous education for detection and the application of risk stratification tools for disease management are crucial. The recently published 2023 WHF guidelines provide a more comprehensive and evidence-based classification of latent RHD, incorporating the hazards for progression into the different diagnostic categories.

## Figures and Tables

**Figure 1 diagnostics-14-00392-f001:**
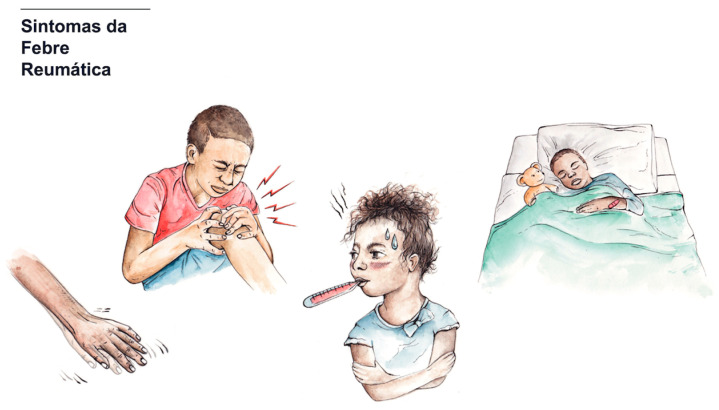
Example of educational flipchart about rheumatic disease utilized for school education.

**Figure 2 diagnostics-14-00392-f002:**
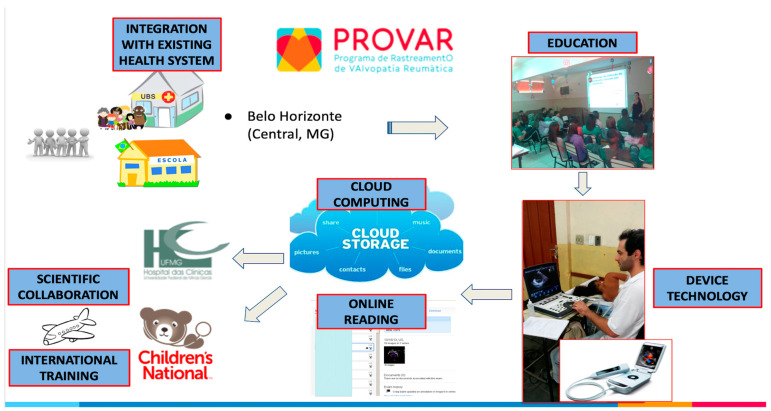
Operational flowchart of the PROVAR+ screening study.

**Table 1 diagnostics-14-00392-t001:** 2012 World Heart Federation Echocardiographic Criteria for Diagnosis of Rheumatic Heart Disease.

**Individuals Aged ≤20 Years**
**Definite RHD (either A, B, C, or D):**
(A) Pathological MR and at least two morphological features of RHD of the MV (B) MS mean gradient ≥4 mmHg * (C) Pathological AR and at least two morphological features of RHD of the AV ‡ (D) Borderline disease of both the AV and MV §
**Borderline RHD (either A, B, or C):**
(A) At least two morphological features of RHD of the MV without pathological MR or MS (B) Pathological MR (C) Pathological AR
**Normal echocardiographic findings (all of A, B, C, and D):**
(A) MR that does not meet all four Doppler echocardiographic criteria (Physiological MR) (B) AR that does not meet all four Doppler echocardiographic criteria (Physiological AR) (C) An isolated morphological feature of RHD of the MV (for example, valvular thickening) without any associated pathological stenosis or regurgitation (D) Morphological feature of RHD of the AV (for example, valvular thickening) without any associated pathological stenosis or regurgitation

**Abbreviations:** AR: aortic regurgitation; AV: aortic valve; MS: mitral stenosis; MR: mitral regurgitation; MV: mitral valve; RHD: Rheumatic Heart Disease. * Congenital MV anomalies must be excluded. Furthermore, inflow obstruction due to nonrheumatic mitral annular calcification must be excluded in adults. ‡ Bicuspid AV, dilated aortic root, and hypertension must be excluded. § Combined AR and MR in high prevalence regions and in the absence of congenital heart disease is regarded as rheumatic.

**Table 2 diagnostics-14-00392-t002:** Consensus diagnosis for latent rheumatic heart disease in screening and follow-up echocardiography, according to the 2012 WHF guidelines, by diagnostic group.

**Final Diagnosis (Follow-Up):**	**Borderline RHD Group (Screening), N = 79**				
	**A**	**B**	**C**	**D**	**Total Borderline**
**Normal**	14 (60.9%)	33 (67.3%)	3 (42.9%)	N/A	50 (63.3%)
**Borderline RHD**	4 (17.4%)	12 (24.5%)	3 (42.9%)	N/A	19 (24.1%)
**Definite RHD**	3 (13.0%)	4 (8.2%)	1 (14.3%)	N/A	8 (10.1%)
**Other**	2 (8.7%)	0	0	N/A	2 (2.5%)
**Final Diagnosis (follow-up):**	**Definite RHD group (screening), N = 16**				
	**A**	**B ***	**C**	**D**	**Total Definite**
**Normal**	0	-	0	1 (50%)	1 (6.3%)
**Borderline RHD**	4 (33.3%)	-	0	0	4 (25%)
**Definite RHD**	8 (66.7%)	-	2 (100%)	1 (50%)	11 (68.8%)
**Other**	0	-	0	0	0

**Abbreviations:** RHD: rheumatic heart disease. * Not available in the absence of spectral Doppler capabilities.

**Table 3 diagnostics-14-00392-t003:** Final classification of screen-positive cases after standard echocardiography.

Category (Screening):	Final Classification (Standard):
**Borderline RHD (N = 79)**	- 19 (24.1%) confirmed as borderline RHD - 50 (63.3%) reclassified to normal (Group A: 14 (28.0%), B: 33 (66.0%), C: 3 (6.0%)) - 8 (10.1%) reclassified to definite RHD (Group A: 3 (37.5%), B: 4 (50.0%), C: 1 (12.5%)) - 2 (2.5%) reclassified as CHD (Group A: 2 (100%))
**Definite RHD (N = 16)**	- 11 (68.8%) confirmed as definite RHD (Group A: 8 (72.7%), C: 2 (18.2%), D: 1 (9.1%)) - 4 (25%) reclassified to borderline RHD (Group A: 4 (100%)) - 1 (6.3%) reclassified to normal (Group D)
**Congenital heart disease (N = 11)**	- 4 (36.4%) confirmed as CHD - 5 (45.5%) reclassified to normal - 2 (18.1%) reclassified to definite RHD (Group A)

**Abbreviations:** CHD: congenital heart disease; RHD: rheumatic heart disease.

**Table 4 diagnostics-14-00392-t004:** Detailed diagnostic information of screening echocardiography, according to the latent rheumatic heart disease category in screening, considering a final positive diagnosis (borderline or definite) in follow-up echocardiography.

Category:	All Screen-Positive (N = 110) *	Definite RHD (N = 16) **
**Sensitivity (%, 95% CI)**	95.5 (84.5–99.4)	34.1 (20.5–49.9)
**Specificity (%, 95% CI)**	19.7 (10.9–31.3)	98.5 (91.8–100.0)
**PPV (%, 95% CI)**	5.3 (4.7–6.0)	10.1 (1.5–45.2)
**NPV (%, 95% CI)**	98.9 (95.6–99.7)	99.7 (99.6–99.7)
**Accuracy (%, 95% CI)**	23.1 (15.6–32.1)	98.2 (93.6–99.8)
**LR+ (N, 95% CI)**	1.19 (1.04–1.36)	22.50 (3.08–164.27)
**LR− (N, 95% CI)**	0.23 (0.05–0.97)	0.67 (0.54–0.83)

**Abbreviations:** LR+: positive likelihood ratio; LR−: negative likelihood ratio; NPV: negative predictive value; PPV: positive predictive value; RHD: rheumatic heart disease. * Considering an estimated prevalence of 4.5%; ** Considering an estimated prevalence of 0.5%.

## Data Availability

The procedures and methods of this study will be made available for replication upon reasonable request directed to the corresponding author.

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
