# Peer review of "Agreement between Handheld and Standard Echocardiography for Diagnosis of Latent Rheumatic Heart Disease in Brazilian Schoolchildren from High-Prevalence Settings (Agreement between Screening and Standard Echo for RHD)†"

_diagnostics, 2024, doi:10.3390/diagnostics14040392_

Round 1

Reviewer 1 Report

Comments and Suggestions for Authors

The study has great potential for clinical utility. Information that is not clear in the manuscript should be completed or clarified. 

1. What were the inclusion and exclusion criteria for the study? Was inclusion based only on the presence of symptoms? What about obese children, for example?  

2. in the case of the children's heart study, the age spread of 11-18 years may prove crucial to the results obtained. I would ask for an analysis with a breakdown that includes pre-pubertal and pubertal subjects.

3. In the study, the RHD criteria for age 20 was used. How does this relate to the age criteria of the children studied, such as 11 years old?

4.How were false-negative cases verified? There is no data on this in the paper. 

5. No information about the Ethics Committee's research approval. The approval number should be given. Only parental consent was mentioned in the paper.  

Comments on the Quality of English Language

Minor editing of English language required.

Author Response

Belo Horizonte, February 6, 2024

Mr. Pawel Zarkowski

MANAGING EDITOR

MDPI Diagnostics

Dear Mr. Zarkowski,

Attached, you will find attached revised a copy of the original article: “Agreement Between Handheld and Standard Echocardiography for Diagnosis of Latent Rheumatic Heart Disease in Brazilian Schoolchildren From High-Prevalence Settingsthat we are submitting for reconsideration in MDPI Diagnostics.

We have worked on the manuscript to improve the minor comments made by the reviewers. Modifications suggested by the reviewer were made and the questions were answered and listed below in this letter. The modifications that we made to the text are indicated in each answer.

The positive comments helped us improve the quality of the report. We hope that the revised manuscript is now acceptable for publications.

Reviewer #1:

The study has great potential for clinical utility. Information that is not clear in the manuscript should be completed or clarified.

A: Thank you for your positive comments and for revising the manuscript. Your suggestions helped us improve the article’s quality. Please see our answers point-by-point below.

  1. What were the inclusion and exclusion criteria for the study? Was inclusion based only on the presence of symptoms? What about obese children, for example?

A: Thank you for this comment. In the 1st paragraph of the Methods section, we made it clearer to the reader that all children from the included schools, aged 11 – 18 years, asymptomatic and without personal history of acute rheumatic fever or rheumatic heart disease were included. These criteria were considered because this was a screening study, aimed at diagnosing subclincal / latent disease. Also, there were no restrictions related to nutritional or pubertal status, and this was also made clearer in this sentence.

  1. In the case of the children's heart study, the age spread of 11-18 years may prove crucial to the results obtained. I would ask for an analysis with a breakdown that includes pre-pubertal and pubertal subjects.

A: We did not systematically collect data about the pubertal status of the included schoolchildren. Thus, the analysis with a breakdown by this status was not possible. In general, the incidence of RHD increases with age, as a cumulative process. However, there is no evidence about a direct impact of puberty on disease severity. Anyway, this was also included as a 5th Limitation in the appropriate section: “Fifth, due to the pragmatic study protocol, breakdown of the analyses by variables as pubertal and nutritional status, or by familial history of RHD, was not possible”.

  1. In the study, the RHD criteria for age 20 was used. How does this relate to the age criteria of the children studied, such as 11 years old?

A: The World Heart Federation criteria (2012) applied for the study considered different criteria for those aged ≤20 years and individuals >20 years. In this study, we utilized the criteria for those ≤20 years, as previously stated in Table 1, and now included in the Methods section of the text (4th paragraph of this section), following your recommendations.

  1. How were false-negative cases verified? There is no data on this in the paper.

A: Unfortunately, the false-negative cases were not verified as, for logistical and cost reasons, only individuals with positive screening echo underwent standard echocardiography. The RHD-negative individuals undergoing standard echo were preliminarily diagnosed with mild congenital heart disease, as explained in the text. We do agree that this is a major limitation of the study, and we included it as the 4th limitation, in the Limitations and Strengths section. Please let us know if any additional statements are necessary.

  1. No information about the Ethics Committee's research approval. The approval number should be given. Only parental consent was mentioned in the paper. 

A: The IRB approval for the study had been previously mentioned in the 1st paragraph of the Methods section, and now we included the approval number. In addition, we made in clearer in the 2nd paragraph of this section that, according to Brazilian research regulations, a signed parental informed consent plus assent forms were required prior to enrollment for those <18 years, and a self-informed consent form was mandatory for those >= 18 years. Please let us know if any additional information is necessary.

We hope that the revised manuscript is now acceptable for publications in MDPI Diagnostics. If you have any additional questions, we would be pleased to provide the answers and explanations as soon as possible.

Once again we would like to thank you and the editorial board for your reconsideration.

I look forward to hearing from you.

Sincerely yours,

Bruno R Nascimento

Bruno Ramos Nascimento, MD, MSc, Ph.D, FACC, FESC

Associate Professor of Medicine

Hospital das Clínicas da Universidade Federal de Minas Gerais

American Heart Association Professional Membership ID: 210403212

Rua Muzambinho 710, apt. 802, Serra

Belo Horizonte, Minas Gerais, Brasil, CEP 30.210-530

Tel.: +55 31 3409 9437; Fax: +55 31 32847298.

Twitter: @ramosnas

Reviewer 2 Report

Comments and Suggestions for Authors

The authors assessed the agreement of RHD screening by non-physicians with handled devices compared to standard fully functional echo exams performed by cardiologists. My comments are as follows:

1.    Significance: Just as the authors stated, the results of the current study apply only to low-resource settings, which limits the novelty generation of the results.

2.    Methods: Basically, this is a diagnostic test. Thus, we expected to find all the essential elements of a diagnostic test in the results. However, the results part is too crude to follow. Selected important indicators for a diagnostic test, for example, ICC was not reported. I suggest the authors refer to another diagnostic test study of the results and indicators.

Comments on the Quality of English Language

None.

Author Response

Belo Horizonte, February 6, 2024

Mr. Pawel Zarkowski

MANAGING EDITOR

MDPI Diagnostics

Dear Mr. Zarkowski,

Attached, you will find attached revised a copy of the original article: “Agreement Between Handheld and Standard Echocardiography for Diagnosis of Latent Rheumatic Heart Disease in Brazilian Schoolchildren From High-Prevalence Settingsthat we are submitting for reconsideration in MDPI Diagnostics.

We have worked on the manuscript to improve the minor comments made by the reviewers. Modifications suggested by the reviewer were made and the questions were answered and listed below in this letter. The modifications that we made to the text are indicated in each answer.

The positive comments helped us improve the quality of the report. We hope that the revised manuscript is now acceptable for publications.

Reviewer #2:

  1. Significance: Just as the authors stated, the results of the current study apply only to low-resource settings, which limits the novelty generation of the results.

A: Thank you for revising the manuscript. Yes, the analyses reflect the findings of low-resourced settings, where RHD prevalence tends to be higher, impacting the results of screening studies. On the other hand, the study may be useful for other programs worldwide aimed at refining screening methodologies for early diagnosis of the disease, once RHD presents a much higher prevalence in undersourced settings, as this is a socially mediated condition, influenced by factors as poor access to healthcare and overcrowding.

  1. Methods: Basically, this is a diagnostic test. Thus, we expected to find all the essential elements of a diagnostic test in the results. However, the results part is too crude to follow. Selected important indicators for a diagnostic test, for example, ICC was not reported. I suggest the authors refer to another diagnostic test study of the results and indicators.

A: In accordance with your recommendation, we included Table 4, with more detailed diagnostic information, such as sensitivity and specificity with 95% confidence intervals, negative and positive predictive values and accuracy. In addition, we added a citation referring to a diagnostic test study, as recommended. Please leu us know if any additional information is required.

We hope that the revised manuscript is now acceptable for publications in MDPI Diagnostics. If you have any additional questions, we would be pleased to provide the answers and explanations as soon as possible.

Once again we would like to thank you and the editorial board for your reconsideration.

I look forward to hearing from you.

Sincerely yours,

Bruno R Nascimento

Bruno Ramos Nascimento, MD, MSc, Ph.D, FACC, FESC

Associate Professor of Medicine

Hospital das Clínicas da Universidade Federal de Minas Gerais

American Heart Association Professional Membership ID: 210403212

Rua Muzambinho 710, apt. 802, Serra

Belo Horizonte, Minas Gerais, Brasil, CEP 30.210-530

Tel.: +55 31 3409 9437; Fax: +55 31 32847298.

Twitter: @ramosnas

Round 2

Reviewer 2 Report

Comments and Suggestions for Authors

None.